

# Psychometric characteristics of the Narcissistic Admiration and Rivalry Questionnaire (NARQ): Arabic version

Aiche Sabah[1], Musheer A. Aljaberi[2], Salima Hamouda[3], Djamila Benamour[1], Keltoum Gadja[1], Yu-Chen Lai[4], Chuan-Yin Fang[4], Amira Mohammed Ali[5] and Chung-Ying Lin[6]

[1] Faculty of Human and Social Sciences, Hassiba Benbouali University of Chlef, Chlef, Algeria
[2] Department of Internal Medicine, Section Nursing Science, Erasmus University Medical Center (Erasmus MC), Rotterdam, Netherlands
[3] Faculty of Human and Social Sciences, University of Mohamed kheider Biskra, Biskra, Algeria
[4] Division of Colon and Rectal Surgery, Ditmanson Medical Foundation Chia-Yi Christian Hospital, Chiayi, Taiwan
[5] Department of Psychiatric Nursing and Mental Health, Faculty of Nursing, Alexandria University, Alexandria, Egypt
[6] Institute of Allied Health Sciences, College of Medicine, National Cheng Kung University, Tainan, Taiwan

## ABSTRACT

Narcissists are characterized by confidence, fragility, a desire for social approval without showing interest in others, charm, self-assurance, arrogance, and aggression. This study assesses the psychometric properties of the Arabic version of the Narcissistic Admiration and Rivalry Questionnaire (NARQ) among Algerian students ($N = 714$). Confirmatory factor and Rasch analyses were used. The NARQ consists of 18 items addressing six narcissism subscales under two main dimensions: rivalry and admiration. The results showed good saturation of the items on the six subscales and the three sub-scales on each of the two main dimensions, revealing a modest but positive correlation between rivalry and admiration. Moreover, the results of the Rasch model demonstrated that the scale aligns with the data, confirming the validity of the scale. This study offers valuable perspectives on assessing narcissism among Arabic populations and enhances our comprehension of the traits linked to narcissistic personalities.

Corresponding authors
Aiche Sabah, s.aiche@univ-chlef.dz
Chuan-Yin Fang, 04969@cych.org.tw

## INTRODUCTION

Over the past few decades, an increase in the prevalence of narcissistic traits among university students has been observed in various studies (*Ahmadi, Nasrolahi & Mirshekar, 2015*; *Pourramzani & Monajemi, 2021*; *Westerman et al., 2011*), characterized by an inflated sense of self-importance and insatiable need for admiration (*Chinnarasri, Wongpakaran & Wongpakaran, 2021*; *Velji & Schermer, 2024*). Ongoing discussions on the conceptualization of narcissism have progressed beyond a singular perspective and now encompass multiple dimensions, including vulnerability and grandiosity (*Ackerman, Donnellan & Wright, 2019*). The correlation between narcissism and health behaviors is complex, particularly

during the COVID-19 pandemic. *Vint et al. (2024)* found that higher narcissism levels were linked to non-adherence to public health measures and anti-vaccination attitudes. *Vaal et al. (2023)* also reported that narcissism and belief in COVID-19 conspiracies predicted reduced compliance with preventive behaviors. *Venema & Pfattheicher (2021)* observed that grandiose narcissism was associated with lower perceived susceptibility to COVID-19, potentially hindering protective measures compliance. Individuals diagnosed with narcissistic personality disorder face an elevated risk of developing organic diseases, displaying poor adherence to therapy, and engaging in detrimental habits (*Perego & Di Mattei, 2020*). The altered stress reactivity linked to narcissistic traits, such as grandiosity and fragility, can significantly impact behavioral, biological, and psychological health outcomes (*Coleman, Pincus & Smyth, 2019*).

Further emphasizing the importance of considering mental and physical health factors within the context of narcissism, previous studies have explored the diverse manifestations of narcissism, ranging from subclinical to clinical manifestations. *Green, MacLean & Charles (2022)* shed light on gender disparities in clinical assessments, where females are underrepresented due to the focus on masculine-aligned and grandiose themes. *Pincus, Cain & Wright (2014)* proposed a clinical model crucial for effective psychotherapy by distinguishing between narcissistic grandiosity and vulnerability. *Sprio et al. (2024)* uncovered a significant link between vulnerable narcissism and self-harm behaviors through a systematic review. Additionally, *Velji & Schermer (2024)* reported mixed associations between narcissistic traits and purpose in life, highlighting the multifaceted nature of narcissism and the necessity for comprehensive understanding across various domains.

Narcissism is a prevalent concept among the general public and research community. Narcissists are often described as confident yet fragile and seeking social approval but displaying little interest in others. They are charming and self-assured but also arrogant and aggressive. Additionally, though they may initially impress their peers, partners, co-workers, and supervisors, they tend to incite relationship conflicts and dissolution in the long term (*Back, 2018*; *Chinnarasri, Wongpakaran & Wongpakaran, 2021*). Recent studies have extensively explored the concept of narcissism. The factorial structure of different measures of narcissism has revealed several aspects. At the same time, previous results have consistently shown that narcissism is associated with both positive attributes, such as charm and cheerfulness, as well as negative behavioral outcomes, including exploitative and manipulative behavior, arrogance, social rejection, conflicts, and decreased life satisfaction (*Edelstein, Newton & Stewart, 2012*; *Fehn & Schütz, 2021*; *Jauk et al., 2021*).

Nevertheless, to elucidate these seemingly contradictory associations and results, several multidimensional models of narcissism have been developed (*Sivanathan, Bizumic & Monaghan, 2021a*). *Krizan & Herlache (2018)* model illustrates narcissism as a self-important trait characterized by an exaggerated sense of worth and importance. They distinguished between two main dimensions: narcissistic grandiosity (arrogance and exhibitionism) and narcissistic vulnerability (dissatisfaction and defensiveness), reflecting a spectrum from subclinical to clinical manifestations. This perspective aligns with the idea that problematic personality traits, like narcissism, exist on a continuum, with most individuals displaying subclinical levels. *Miller et al. (2016)* supported the

multidimensional nature of narcissism by identifying three narcissism constructs: antagonism (*e.g.*, arrogance), neuroticism (*e.g.*, the need for admiration), and agentic extraversion (*e.g.*, authority). Despite variations, they offered similar definitions, suggesting a spectrum from subclinical to clinical manifestations. Additionally, *Miller et al.*'s (*2016*) model distinguished between grandiose and vulnerable narcissism, with subclinical shifts involving varying degrees and clinical shifts indicating a more extreme pattern meeting the criteria for narcissistic personality disorder. Both models hypothesized that grandiose and vulnerable narcissism form the common nucleus, with proposed divisions including intrapersonal and interpersonal narcissism, and covert and overt narcissism. However, *Back et al. (2013)* articulated the concepts of admiration and narcissistic rivalry, deconstructed grandiose narcissism's aspects, which are widely accepted and investigated, and provided a comprehensive understanding of narcissistic traits (*Sivanathan, Bizumic & Monaghan, 2021b*). While all models agree on narcissism's multidimensional nature, they differ in focus: *Krizan & Herlache (2018)* focused on functional modes, *Miller et al. (2016)* on personality traits, and *Back et al. (2013)* on interpersonal dynamics and emotional motivations. *Back et al. (2013)*'s model is particularly noteworthy for its emphasis on interpersonal dynamics, detailed analysis of grandiose narcissism, broad acceptance in research circles, and its valuable framework for understanding and managing narcissistic behavior.

## The narcissistic admiration and rivalry concept

Grandiose or agentic narcissism involves the comprehensive self-evaluations of grandiose narcissists, who exhibit an excessive sense of self-importance, entitlement, and social power. Agentic narcissists emphasize attributes like intelligence, creativity, and competence while disregarding communal qualities such as agreeableness, fairness, and cooperation (*Nehrlich et al., 2019*). Moreover, researchers have taken great steps to solve the complexity of narcissism (*Kirk et al., 2022*), and their contributions have produced several models. Perhaps the most prominent of these is the model of *Back et al. (2013)*, which defined two distinct forms of narcissism, admiration and rivalry, to develop the Narcissistic Admiration and Rivalry Questionnaire (NARQ). According to the NARQ framework, narcissism consists of two separate dimensions, both of which help narcissistic individuals maintain their great self-perceptions. However, each dimension includes unique cognitive, motivational, and behavioral processes (*Seidman, Shrout & Zeigler-Hill, 2020*).

Admiration is related to the strategy of preserving grandness, which refers to a defensive and inflated self-image that individuals develop as a response to their natural narcissism being inevitably challenged by occasional failures and inadequate responses from others. This preservation of grandness is achieved by obtaining the admiration of others (assertive self-enhancement). This strategy is linked to emphasizing the uniqueness and specialization of the individual, along with fantasies about their grandiosity and charming behavior that can lead to positive social outcomes. Consequently, these positive social encounters fuel the grandiose self and further reinforce the assertive self-enhancement approach (*Colman, 2009*; *Mück et al., 2020*). On the other hand, narcissistic rivalry describes a defensive personal strategy based on deployed efforts to protect the great self-views by reducing the value of others and depreciating them to feel supereminence. Such efforts lead to aggressive

behavior and social conflict, which in turn threaten the narcissist's ego and lead to an increase in self-defense strategies (*Fehn & Schütz, 2021*). Moreover, narcissistic admiration and narcissistic rivalry have moderate to strong levels of association, meaning that the two dimensions can exist together, but not always (*Gauglitz et al., 2022*).

The NARQ has shown acceptable evidence of validity and reliability across diverse populations and contexts. *Back et al. (2013)* distinguished between the admiration (firm self-reinforcement) and rivalry (hostile self-protection) strategies of narcissism through seven validation studies. The NARQ has since been translated into various languages and exhibited promising psychometric properties in different populations (*Doroszuk et al., 2020*; *Jota, 2021*; *Vecchione et al., 2018*). Its validity and reliability have been firmly established in Spanish-speaking cultures (*Doroszuk et al., 2020*), with confirmatory factor analysis supporting the two-factor structure and sufficient internal consistency levels in the Italian environment (*Vecchione et al., 2018*). In the Venezuelan population, the NARQ demonstrated good psychometric characteristics and corroborated the literature by linking admiration to personal dominance, while rivalry reflected low collective orientation or hostility. A robust association was also observed between admiration/rivalry and constructs like self-esteem, cheerfulness, openness, and agreeableness (*Jota, 2021*). *Leckelt et al. (2018)* provided further validation by utilizing large convenience ($n = 11,937$) and representative ($n = 4,433$) samples, confirming the robust factor structure of the NARQ-Short Scale (NARQ-S) and establishing it as a reliable and valid measure of admiration and rivalry narcissistic traits. Notably, the NARQ has proven its applicability in clinical contexts as well. *Mota et al. (2019)* investigated the relationship between admiration/rivalry preferences and mental disorders/medication usage in the general population ($n = 2,513$) and clinical samples ($n = 475$). Their study successfully validated the two-dimensional structure and reliability of the NARQ within clinical samples. Overall, the cumulative evidence across these studies firmly supports the NARQ as a valid and reliable instrument for assessing narcissistic admiration and rivalry in both general and clinical populations across diverse cultural contexts.

## Literature gap and the purpose of the present study

The studies mentioned above highlight the importance of the NARQ in measuring narcissism and its global interest, as it has been translated into multiple languages and consistently demonstrated good levels of validity and reliability. This reliability has encouraged researchers to validate the psychometric characteristics of the Arabic version of the scale, specifically on Algerian students. There is a literature gap regarding the evidence of psychometric properties for the Arabic version of the NARQ. Arabic cultures have shifted from collectivist to individualist cultures, which may result in changes in narcissism among Arabic individuals (*Lyons et al., 2013*). Prior evidence shows that collectivist individuals, compared to individualist individuals, score higher narcissistic scores (*Foster, Keith Campbell & Twenge, 2003*). However, without a validated instrument, it is hard to determine the narcissistic characteristics of Arabic individuals.

Using a validated instrument that assesses narcissistic characteristics (*i.e.,* the NARQ) allows authorities and relevant stakeholders to identify the features of narcissism among

Arabic individuals. Accordingly, the NARQ could help monitor whether individuals have low levels of narcissism that need further intervention. Subsequently, health may be improved for those with low levels of narcissism.

This study aims to examine the validity and reliability of a tool (*i.e.*, NARQ) for measuring narcissism in the Algerian environment. We assessed its psychometric characteristics using confirmatory factor analysis (CFA) and the Rasch model. The study objectives are twofold: (1) to evaluate the psychometric characteristics of the Arabic version of the NARQ using CFA, and (2) to evaluate the psychometric characteristics of the scale using the Rasch model.

## METHODS

### Study design, sample, and recruitment procedure

The study was conducted by implementing a cross-sectional study design. The study adhered to the ICMJE guidelines, Declaration of Helsinki, and STROBE checklist. The Department of Social Sciences, Faculty of Humanities and Social Sciences, Hassiba Ben Bouali University, Chlef, Algeria, granted Ethical approval to conduct the study within its facilities (Ethical Application Ref: 2023/gandm/19). Written informed consent was obtained from all study participants for their voluntary participation and consent to publish study findings. The study participants were 714 university students (12% male; 88% female) from Chlef University, Saida, and Biskra in Algeria. The participants' ages ranged from 17 to 54 years, with an average age of 22.19 and a standard deviation of 4.09. Table 1 shows the participant characteristics: marital status, family economic level, students' specialties, and academic year.

Before collecting data from the university students, the researchers translated the NARQ from English to Arabic. The authors have permission from the copyright holders to use this instrument. The translation process adhered to the guidelines provided by the International Test Commission for Translating and Adapting Tests throughout the research process (*Commission, 2017*). Regarding the questionnaire application, it was conducted manually in study departments within the faculties. Contact was made, and assistance was sought from certain faculty members who generously allocated time for us to administer the questionnaire. The purpose of the questionnaire was explained, emphasizing its voluntary nature and sole use for scientific research purposes. The questionnaire was administered to 714 students, and the sample size was deemed sufficient to calculate the questionnaire's psychometric properties.

### Sample size

Multiple researchers have offered guidance on adequate sample sizes in CFA and structural equation modeling (SEM). *Boomsma & Hoogland (2001)* recommended a minimum sample size of 200 for simple models and at least 400 for more complex ones. *Kline (2023)* categorized sample sizes as small (under 100), medium (100–200), and large (over 200), which are generally acceptable for most models. While researchers concur that larger sample sizes are preferable for SEM/CFA, specific recommendations differ. A general consensus is that a sample size of N>=200 is suitable for most models. However, smaller samples

**Table 1** **The social and demographic characteristics of the participants.** The table provides an overview of the social and demographic characteristics of the participants, such as gender, marital status, family economic situation, academic field, and current academic level. The majority of participants are female and unmarried. Most come from families with moderate economic status, and they pursue diverse academic interests at different stages of their education.

| Variables | Groups | n | Percentage |
|---|---|---|---|
| Gender | Male | 86 | 12.0 |
| | Female | 628 | 88.0 |
| Marital status | Single | 650 | 91.0 |
| | Married | 64 | 9.0 |
| Economic status of families | Lower | 22 | 3.1 |
| | Middle | 598 | 83.8 |
| | Upper | 94 | 13.1 |
| Students' specialty | Medicine and Natural Sciences | 15 | 2.1 |
| | Social and human Sciences | 669 | 93.7 |
| | Languages | 30 | 4.2 |
| Students' academic year | first year | 99 | 13.9 |
| | Second year | 172 | 24.1 |
| | Third year | 120 | 16.8 |
| | Master 1 | 71 | 9.9 |
| | Master 2 | 244 | 34.2 |
| | Doctorate | 8 | 1.1 |

may be adequate for simpler models with normal data and numerous indicators per factor. Ideally, the ratio of cases to estimate parameters (N:q) should be 10:1 or higher, although 5:1 may suffice in certain circumstances (*Alareqe et al., 2022*; *Harrington, 2009*; *Sabah et al., 2023*; *Wang & Wang, 2019*). Furthermore, according to *Hair et al. (2019a)*; *Hair et al. (2019a)* sample size over 100 is required when using SEM for covariance. Generally, larger samples are preferred, considering estimation methods, model complexity, and data characteristics. For this study, a sample size of 714 was deemed appropriate and adequate to employ SEM/CFA and achieve the research objectives.

## Instrument

The NARQ (*Back et al., 2013*) proposes a model that includes two dimensions of narcissism: admiration and rivalry. The scale suggests that narcissists pursue the comprehensive goal of preserving their great self through two tracks, each characterized by distinct cognitive, emotional, and behavioral processes. The 18-item NARQ provides six graded response options (ranging from 1 (do not agree at all) to 6 (completely agree)) to evaluate nine phrases for each of the two narcissistic strategies: admiration and rivalry. The model of narcissism distinguishes admiration and rivalry based on three key dynamics: behavioral (charming *vs.* aggressive behavior), emotional-motivational (striving for uniqueness *vs.* supremacy), and cognitive (grandiose fantasies *vs.* devaluation of others). Admiration involves seeking social approval through assertive self-enhancement, while rivalry focuses on preventing social failure through antagonistic self-protection. These dimensions operate

through distinct cognitive, emotional-motivational, and behavioral processes, influencing interpersonal outcomes such as social potency and conflict.

We translated the NARQ into Arabic using the guidelines of the International Test Commission for Translating and Adapting Tests (*Commission, 2017*). First, we obtained permission from the scale owner (Prof. Dr. Mitja Back), who allowed us to translate it into Arabic. Before translating the scale, we evaluated the item content, scale structure, and compatibility with the Arab environment where the scale would be implemented. We found that the elements and scale structure were clear, and none contradicted Arab culture. A translator fluent in English and the target language with in-depth knowledge of English culture was asked to translate the scale from English into Arabic. The translator was based in Algeria. After that, an English translation specialist performed a back translation on the scale. This specialist is proficient in Arabic and English and has translated it into English. The translation was reviewed by researchers who were more familiar with the test content and assessment principles, and the translation process was discussed with the research team regarding its linguistic suitability. Then, the scale was presented to a group of university students as the target audience to check the readability. The university students confirmed their understanding of the items and the absence of ambiguity. Therefore, the NARQ is finalized for further psychometric testing (*i.e.,* the CFA and Rasch analysis).

## Data analysis

The data were analyzed using SPSS 26 for preliminary analysis, including descriptive statistics, Cronbach's alpha, and McDonald's omega. AMOS 24 was employed for CFA, while Winsteps 3.72.3 was used for Rasch analysis. Moreover, all the materials associated with the present study's analyses are provided in the supplementary material.

CFA is pivotal for assessing the validity of hypotheses concerning the relationships between latent and observed variables (*Aljaberi et al., 2023*; *Aljaberi et al., 2018*; *Sabah & Al-Shujairi, 2022*). Thus, CFA was employed to validate whether our study's data structure aligned with the previously outlined factor structure of the NARQ in its original version. Specifically, the NARQ was anticipated to exhibit a two-dimensional structure with three subscales within each factor. CFA utilized the maximum likelihood estimation method. Several indices, including normed chi-square (*i.e.,* chi-square divided by degrees of freedom), comparative fit index (CFI), Tucker-Lewis index (TLI), root mean square error of approximation (RMSEA), and standardized root mean square residual (SRMR), were evaluated to assess model fit. A normed chi-square below 5 indicates an acceptable level of fit (*Abiddine et al., 2022*; *Aljaberi et al., 2022a*; *Fares et al., 2021*; *Marsh & Hocevar, 1985*; *Munro, 2005*; *Sabah, 2019*), CFI and TLI greater than 0.90 signifies an acceptable fit of the data with the model, RMSEA below 0.08 demonstrates an acceptable degree of fit and above 0.10 suggests a poor fit, SRMR lower than 0.08 indicate superior fit (*Beck, 2013*; *Byrne, 2013*; *Ntoumanis & Myers, 2015*; *Sabah, 2019*; *Sabah et al., 2022*; *Sabah & Al-Shujairi, 2022*; *Sabah, Khalaf Rashid Al-Shujairi & Boumediene, 2021*; *Sürücü, Şeşen & Maslakçı, 2023*).

Using the coefficients obtained from the CFA, the model's validity was further examined using the following statistics. These statistics included composite reliability (CR), average variance extracted (AVE), maximum reliability (Max H), and heterotrait-monotrait ratio

(HTMT) (*Alareqe et al., 2022*; *Aljaberi et al., 2022b*). According to *Hair, Howard & Nitzl (2020)*, CR values ideally fall between 0.70 and 0.95, and AVE values should be higher than 0.5. Additionally, the criterion for discriminant validity was satisfied when the AVE values were greater than the maximum shared variance (MSV) values (*Kumar & Singh, 2021*). For Max H, values greater than 0.7 are acceptable (*Rahmatpour, Peyrovi & Nia, 2021*). The HTMT is also used to assess discriminant validity. *Henseler, Ringle & Sarstedt (2015)* state that all HTMT ratios should be lower than 0.85 to achieve satisfactory discriminant validity.

The psychometric properties of the NARQ were then assessed using the one-parameter (*i.e.,* difficulty) Rasch model. The data underwent Rasch analysis using the Winsteps program (*Linacre, 2022*), which employed the Rasch rating scale model for analyzing polytomous data. In the Rasch analysis, we calculated fit statistics to evaluate the unidimensionality requirement of the Rasch model. This included weighted and unweighted mean squares of residuals (Infit and Outfit MnSqs) (*Abiddine et al., 2024*). For assessing the functioning of the NARQ response categories, mean square values for Infit and Outfit MnSq within the range of 0.6 to 1.4 are considered productive for rating scale measurement (*Bond & Fox, 2015*). Additionally, as noted by *Bond & Fox (2013)*, a monotonic increase in the average of the measures indicates that individuals with higher abilities endorse progressively higher categories, while those with lower abilities endorse progressively lower categories.

The Rasch model illustrates the distribution through the Wright Map, which visually displays the connection between individuals and items on a logit scale. Items are arranged from the most difficult to the easiest on the right side, while individuals are plotted from the highest ability to the lowest on the left side. This map showcases whether items span the entire range of the construct and match well with individuals' abilities. It assesses the quality of the instrument, its targeting, and theoretical alignment (*Boone, 2016*; *Boone & Noltemeyer, 2017*). Item separation and reliability are important aspects of measurement accuracy. Separation refers to how well items or people can be differentiated based on the trait being measured. It shows how item difficulties, or a person's abilities, are spread out on the measurement scale.

On the other hand, reliability indicates how consistent or stable the measurements are in reflecting the underlying trait. When item separation is greater than 3.0 and reliability is higher than 0.90, it suggests that the estimates of item difficulty are stable. Similarly, when a person's separation exceeds 2.0, and reliability is above 0.80, stable estimates of a person's ability or trait level are obtained (*Van Zile-Tamsen, 2017*).

## RESULTS

### Item properties and CFA results of the narcissistic admiration and rivalry questionnaire

Table 2 shows that the arithmetic average of items ranged between 1.81 (RivDe2) "Other people are worth nothing," and 4.72 (AdmUn2) "I enjoy my successes very much". Kurtosis ranged between −1.00 to 1.29, and the skewness values ranged between −0.71 to 1.43. Regarding items that exceed one, there are only two items (RivDe2, and RivSu3).

**Table 2 Item and factor analysis of the Narcissistic Admiration and Rivalry Questionnaire (NARQ).** The table provides a thorough analysis of the Narcissistic Admiration and Rivalry Questionnaire (NARQ), scrutinizing its items and factors using statistical measures such as skewness, kurtosis, mean, and standard deviation. Validity is confirmed through Cronbach's alpha and omega ($\omega$) across sub-dimensions and the sample. The results showcase diverse responses, with skewness and kurtosis statistics reinforcing the questionnaire's reliability, thus emphasizing the NARQ's efficacy in capturing narcissistic traits.

| Variables | | | Items | M | SD | Skewness | kurtosis | Factor loadings | $\alpha$ | Omega ($\omega$) |
|---|---|---|---|---|---|---|---|---|---|---|
| NARQ admiration scale | 1. NARQ grandiosity facet | AdmGr1 | I am great | 4.21 | 1.31 | −0.50 | −0.20 | 0.47 | | |
| | | AdmGr2 | I will someday be famous. | 3.50 | 1.46 | −0.02 | −0.84 | 0.56 | 0.59 | 0.61 |
| | | AdmGr3 | I deserve to be seen as a great personality | 3.79 | 1.42 | −0.12 | −0.82 | 0.68 | | |
| | 2. NARQ uniqueness facet | AdmUn1 | I show others how special I am. | 3.67 | 1.46 | −0.09 | −0.87 | 0.66 | | |
| | | AdmUn2 | I enjoy my successes very much | 4.72 | 1.23 | −0.71 | −0.31 | 0.65 | 0.71 | 0.72 |
| | | AdmUn3 | Being a very special person gives me a lot of strength. | 4.50 | 1.27 | −0.59 | −0.33 | 0.71 | | |
| | 3.NARQ charmingness facet | AdmCh1 | Most of the time I am able to draw people's attention to myself in conversations. | 3.81 | 1.36 | −0.18 | −0.66 | 0.74 | | |
| | | AdmCh2 | I manage to be the centre of attention with my outstanding contributions. | 3.73 | 1.25 | −0.19 | −0.43 | 0.76 | 0.77 | 0.77 |
| | | AdmCh3 | Mostly, I am very adept at dealing with other people. | 4.01 | 1.29 | −0.33 | −0.39 | 0.70 | | |
| NARQ rivalry scale | 4. NARQ devaluation facet | RivDe1 | Most people won't achieve anything. | 2.47 | 1.40 | 0.72 | −0.34 | 0.66 | | |
| | | RivDe2 | Other people are worth nothing. | 1.81 | 1.14 | 1.43 | 1.43 | 0.70 | 0.76 | 0.76 |
| | | RivDe3 | Most people are somehow losers | 2.26 | 1.3 | 0.80 | −0.24 | 0.81 | | |
| | 5.NARQ supremacy facet | RivSu1 | Secretly take pleasure in the failure of my rivals. | 2.32 | 1.48 | 0.92 | −0.22 | 0.77 | | |
| | | RivSu2 | I want my rivals to fail. | 2.35 | 1.50 | 0.92 | −0.21 | 0.83 | 0.84 | 0.84 |
| | | RivSu3 | I enjoy it when another person is inferior to me. | 1.98 | 1.31 | 1.41 | 1.29 | 0.76 | | |
| | 6. NARQ aggressiveness facet | RivAg1 | I react annoyed if another person steals the show from me. | 2.42 | 1.47 | 0.81 | −0.37 | 0.78 | | |
| | | RivAg2 | I often get annoyed when I am criticized. | 3.32 | 1.55 | 0.08 | −1.00 | 0.49 | 0.64 | 0.64 |
| | | RivAg3 | I can barely stand it if another person is at the centre of events. | 2.62 | 1.43 | 0.56 | 0.18 | 0.59 | | |
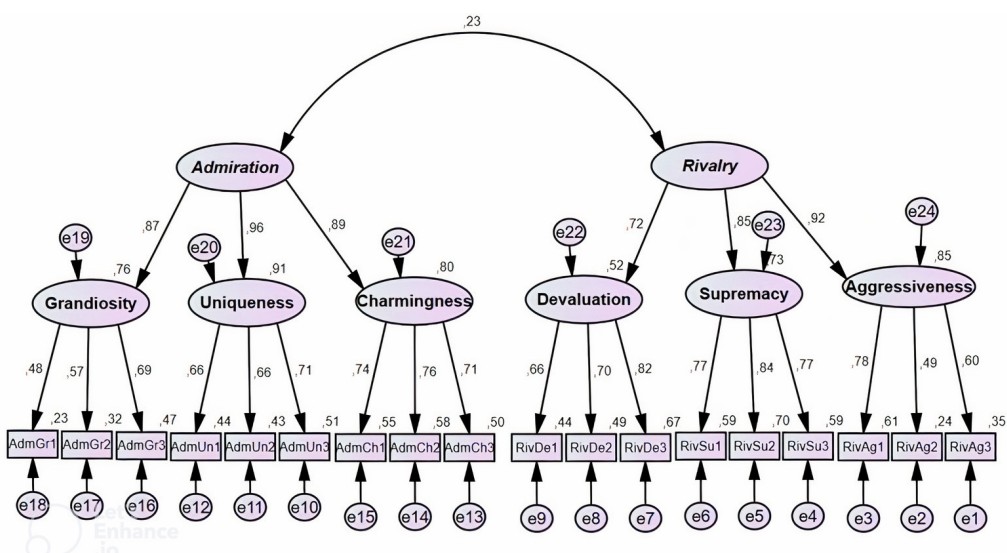

**Figure 1 CFA factor loadings of the NARQ.** This figure depicts the loading of the Narcissistic Admiration and Rivalry Questionnaire (NARQ) through confirmatory factor analysis (CFA) among 714 university students. Model fit measures validate the model's conformity with the data, affirming its acceptable loading.

Furthermore, we noted that the values were limited between −2 and +2, which indicates the normal distribution of the scale items. Cronbach's Alpha values for subscales showed acceptable values (0.64 to 0.84), except for the subscale of grandiosity (0.59). Similar findings were observed in omega ($\omega$) values (0.61 to 0.84).

## Confirmatory factor analysis

The fit indices of the proposed model were normed chi-square (4.48), CFI (0.90), TLI (0.89), RMSEA (0.07), and SRMR (0.07). All other fit indices support a good fit except for the TLI (0.89), which is close to acceptable (0.90). *Xia & Yang (2019)* suggest that the unweighted least squares (ULS) and diagonally weighted least squares (DWLS) estimation methods yield higher TLI values compared to maximum likelihood (ML). This difference can lead to overly optimistic assessments of model fit when conventional cutoff values are applied to ULS and DWLS indices. Consequently, accepting a TLI value below 0.90 depends on the specific context and estimation method used. In our current study, AMOS was employed with the ML method, resulting in a TLI estimate of 0.89—just below the desired threshold of 0.90. This slight variance may be attributed to the use of ML estimation.

Additionally, Fig. 1 displays the factor loadings of the NARQ items. The factor loading values ranged between 0.47 and 0.83, statistically significant at a significance level 0.01. When all items are statistically significant, but some items have loadings less than 0.70—which is considered the threshold value for item loading onto factors, according to *Hair Jr et al. (2021)*—researchers have suggested a minimum loading value of 0.40 (*Stevens, 2009*).

Nonetheless, item loadings between 0.40 and 0.70 can be justified if acceptable values are obtained for other indicators of internal consistency reliability, AVE, and HTMT,

which have been found acceptable as reliability indicators (*Hair et al., 2019b*; *Hair Jr et al., 2021*). Items with loadings below 0.70 were retained due to their theoretical importance in interpreting the latent variable, as they are theoretically crucial and cannot be removed despite their lower loadings. In the grandiosity subscale, items AdmGr1 ("I am great") and AdmGr2 ("I will someday be famous") have loadings of 0.47 and 0.56, respectively. These items are essential for capturing the essence of grandiosity. "I am great" (loading 0.47) embodies the immediate, present-oriented self-aggrandizement, a key trait of narcissistic grandiosity. Without this item, the scale would lose a critical dimension of how grandiosity is expressed and experienced. Its straightforward nature clearly indicates the respondent's grandiose self-view, which is vital for a comprehensive assessment. Similarly, "I will someday be famous" (loading 0.56) captures grandiosity's aspirational and future-oriented aspect. This belief in future fame reflects a strong sense of personal destiny and exceptionalism, core traits of narcissistic grandiosity. Including this item broadens the grandiosity construct by acknowledging both current self-views and future expectations of greatness. Additionally, the minimum number of items in the latent variable should not be lower than three indicators or items when testing a one-factor structure model (*Bollen, 1989*), and deleting any item would result in the latent variable containing only two items, which disrupts its theoretical component and alters its property (*Raubenheimer, 2004*). These items also provide beneficial information about the structure and their retention.

Table 3 additionally illustrates other validity statistics, which also support the promising psychometric properties of the NARQ. Specifically, it is observed that CR was 0.93 in the admiration construct and 0.87 in the rivalry construct, and AVE was 0.82 in the admiration construct and 0.69 in the rivalry construct. Furthermore, the MSV values for both constructs were 0.05, indicating that the criterion for discriminant validity (AVE >MSV) was met. The Max H values were 0.94 for admiration and 0.90 for rivalry, indicating high accuracy and consistency in the model. Regarding discriminant validity, the HTMT was 0.22. The current study found that the stability was less than 0.70 for the "Grandiosity, Aggressiveness" subscales. This can be attributed to *Gudmundsson (2009)* finding that, in most cases, the total scores carry more weight than subscale scores in interpreting results. Moreover, in any translation or adaptation project of a scale to another context, lower reliability coefficients should be expected for subscales and composite measures of the instrument in the target language compared to the original language. In this regard, addressing items with low loadings is essential to improve the measure. For example, the translation could be revised for poorly performing items, especially if these items are truly core to the construct. Therefore, revisiting and refining these items is necessary to ensure the measure's validity and usefulness. In this direction, the NARQ translated into Arabic for the present study should be refined in future research. The present study provides a starting point for the Arabic version of the NARQ, allowing future studies to investigate further and improve the measure.

### Rasch analysis

Figure 2 shows the item person map displaying the distribution of personal values and item difficulty estimates. The item distribution (Fig. 2, Table 4) (mean = 0.00; SD =

**Table 3  Structural validity analysis.** The model's validity was rigorously assessed using various metrics such as Composite Reliability (CR), Average Variance Extracted (AVE), Maximum Shared Squared Variance (MSV), Maximal Reliability MaxR(H), and Heterotrait-Monotrait (HTMT). Results demonstrate acceptable internal consistency, minimal shared variance, high reliability, and discriminant validity, affirming the model's validity

| | CR | AVE | MSV | MaxR(H) | HTMT analysis | |
| | | | | | Admiration | Rivalry |
|---|---|---|---|---|---|---|
| Admiration | 0.93 | 0.82 | 0.05 | 0.94 | 0.90 | |
| Rivalry | 0.87 | 0.69 | 0.05 | 0.90 | 0.22[***] | 0.83 |

Notes.

CR, Composite Reliability; AVE, Average Variance Extracted; MSV, Maximum Shared Variance; Maxr(H), Maximum Reliability; HTM, Heterotrait-Monotrait Ratio (HTMT).

[***] $p < 0.001$.

**Table 4  Item statistics in Rasch analysis.** The Rasch analysis evaluated the fit of the NARQ. Reliability estimates indicate model validity, with item and person classification fitting well. The table details item statistics, affirming the model's adequacy in measuring narcissistic traits.

| NAME | Difficulty | SE of difficulty | IN.MNSQ | IN.ZSTD | OUT.MNSQ | OUT.ZSTD |
|---|---|---|---|---|---|---|
| ADMGR1 | −0.68 | 0.03 | 1.12 | 2.46 | 1.42 | 7.31 |
| ADMGR2 | −0.21 | 0.03 | 1.06 | 1.31 | 1.13 | 2.80 |
| ADMGR3 | −0.40 | 0.03 | 0.86 | −2.96 | 0.87 | −2.73 |
| ADMUN1 | −0.32 | 0.03 | 0.79 | −4.72 | 0.80 | −4.20 |
| ADMUN2 | −1.08 | 0.04 | 1.09 | 1.72 | 1.15 | 2.57 |
| ADMUN3 | −0.90 | 0.03 | 0.90 | −1.91 | 0.90 | −1.70 |
| ADMCH1 | −0.41 | 0.03 | 0.83 | −3.57 | 0.86 | −2.80 |
| ADMCH2 | −0.36 | 0.03 | 0.67 | −7.89 | 0.68 | −7.27 |
| ADMCH3 | −0.54 | 0.03 | 0.82 | −3.71 | 0.84 | −3.17 |
| RIVDE1 | 0.46 | 0.03 | 1.08 | 1.65 | 1.08 | 1.47 |
| RIVDE2 | 1.05 | 0.04 | 1.17 | 2.68 | 1.14 | 1.93 |
| RIVDE3 | 0.62 | 0.03 | 1.04 | 0.85 | 1.03 | 0.62 |
| RIVSU1 | 0.58 | 0.03 | 1.18 | 3.51 | 1.18 | 3.05 |
| RIVSU2 | 0.56 | 0.03 | 1.21 | 3.93 | 1.20 | 3.45 |
| RIVSU3 | 0.87 | 0.04 | 1.11 | 1.94 | 1.06 | 1.00 |
| RIVAG1 | 0.50 | 0.03 | 1.03 | 0.76 | 1.01 | 0.34 |
| RIVAG2 | −0.10 | 0.03 | 1.17 | 3.56 | 1.18 | 3.79 |
| RIVAG3 | 0.36 | 0.03 | 1.03 | 0.65 | 1.03 | 0.68 |

Notes.

SE, standard error; IN.MNSQ, infit mean square; IN.ZSTD, $z$ value of the infit mean square; OUT.MNSQ, outfit mean square; OUT.ZSTD, $z$ value of the outfit mean square.

0.62) extends from the easiest item (ADMUN5; −1.08 to the most difficult to endorse item [RIVDE2]; 1.05). The person distribution (mean = −0.24; SD = 0.57) locates the respondent with the lowest NARQ score (675) is located at −1.77 logits and with the highest narcissism rating (576) at 5.07 logits. Given that the item and person mean and item person distributions were quite similar, it is reasonable to conclude that the narcissism reported by these university students is well targeted by the NARQ Arabic version.

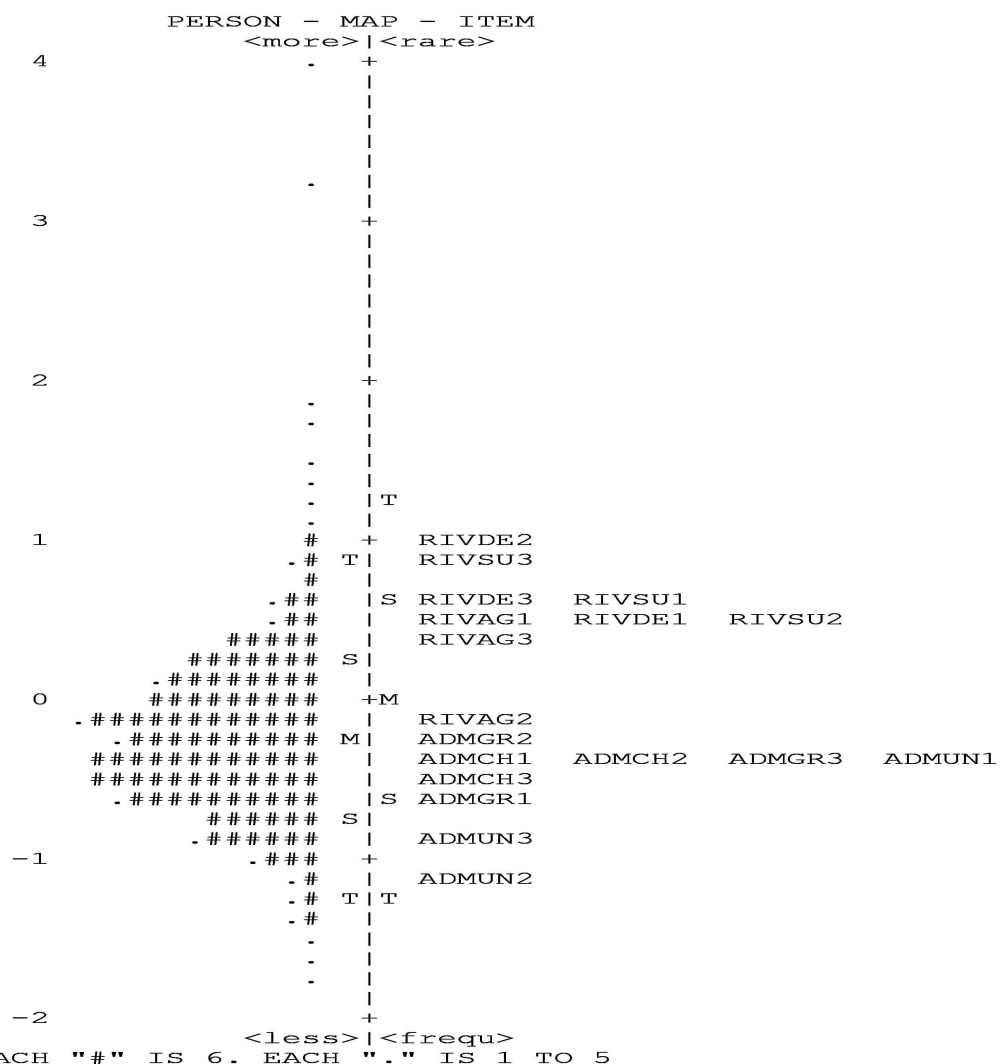

**Figure 2 The NARQ person-item Wright map.** The NARQ item map elucidates the alignment of item difficulty and individual capabilities according to Rasch's analysis. It delineates a hierarchy of difficulty, with the most challenging items positioned at the summit and the less difficult ones below, concurrently illustrating the existence of the floor effect.

The fit of the Rasch model's unidimensionality requirement was further examined. Although item 8 (infit MnSq = 0.67) and item 1 (outfit MnSq = 1.43) did not have satisfactory fit, all other items had satisfactory fit statistics regarding infit MnSq and outfit MnSq. Regarding reliability estimates, the Rasch model results showed that person separation reliability was 0.82, item separation reliability was 1.00, person separation index was 2.16, and item separation index was 18.25. This indicates that the NARQ items and the present sample were coherent, and the NARQ could efficiently classify the items and persons into different groups.

Table 5 summarizes the category functioning analysis under Rasch measurement requirements: no infit and outfit MnSq value exceeded 1.40 or fell below 0.60. Also,

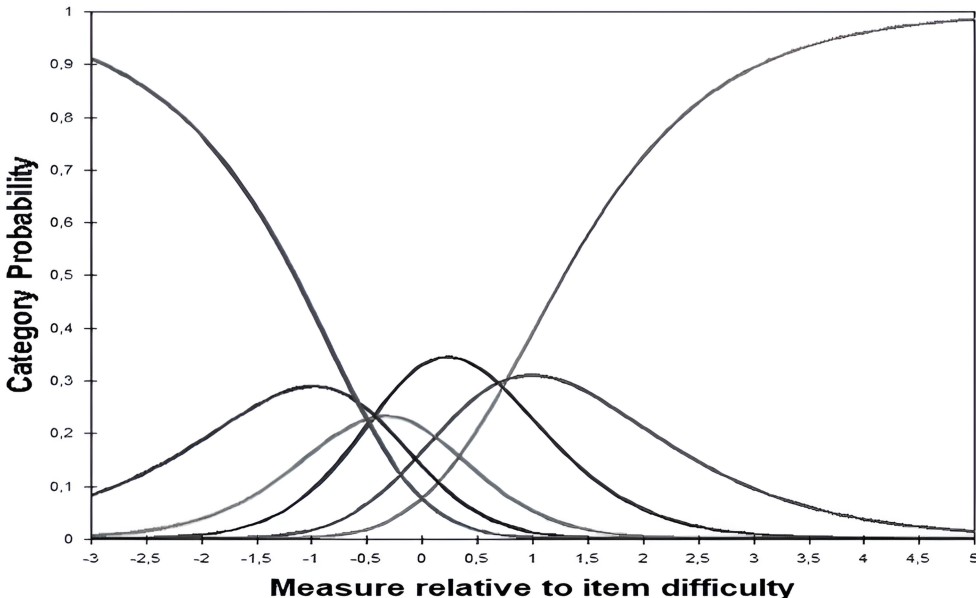

**Figure 3** **Measure relative to item difficulty: six categories.** The output table generated by Winsteps is utilized to assess the Category function analysis. Figure 2's Winsteps result delves into the structure of rating scales, examining the category structure of the instrument. Here, the category probability curves for the Narcissistic Admiration and Rivalry Questionnaire (NARQ) are illustrated, providing insights into the functioning of its rating scale categories.

**Table 5** **Summary of category structure.** Table outlines the analysis of category functioning, adhering to Rasch measurement standards. It guarantees data consistency by maintaining infit and outfit MnSq values within specified limits. Moreover, it highlights a consistent escalation in difficulty across category measures. Enter 20:31.

| Category LABEL | Infit MNSQ | Outfit MNSQ | Structure CALIBRATN | Category MEASURE |
|---|---|---|---|---|
| 1 (not agree at all) | 0.94 | 1.00 | NONE | (−2.13) |
| 2 | 0.99 | 1.02 | −0.59 | −0.98 |
| 3 | 0.99 | 0.96 | −0.42 | −0.33 |
| 4 | 1.08 | 1.17 | −0.44 | −0.23 |
| 5 | 0.91 | 0.91 | −0.69 | −0.99 |
| 6 (agree completely) | 1.11 | 1.12 | 0.77 | −(2. 29) |

**Notes.**
    *MNSQ=mean square.

the monotonically increased difficulty was observed in the category measure. Figure 3 depicts the category probability curves for the NARQ (NARQ 1 (not agree at all), 6 (agree completely)).

## DISCUSSION

The objective of the present study was to examine the psychometric characteristics of the Arabic version of NARQ using a sample of Algerian participants. The findings revealed that the current model aligns with the original scale structure. The study results showed that the hypothetical narcissism model had acceptable quality indicators. This is in line with earlier research where CFA consistently validated the two-dimensional structure of the population and encompassed factors related to admiration and rivalry across both convenience and representative samples (*Back et al., 2013*; *Jota, 2021*; *Leckelt et al., 2018*; *Mota et al., 2019*; *Vecchione et al., 2018*). As for the scale components, the subscales loadings were saturated on the first main dimension, rivalry, with new saturations, whereas the saturations of aggressiveness, supremacy, and devaluation reached 0.92, 0.85, and 0.72, respectively. However, regarding the components of the second main dimension, admiration, the sub-dimensions showed good correlations, as loading saturations of charm, uniqueness, and grandiosity reached 0.89, 0.96, and 0.87, respectively. This indicates the existence of strong correlations between the sub-dimensions and the main dimensions, which is consistent with the findings of the study conducted by *Back et al. (2013)*.

The NARC delves into agentic and antagonistic aspects of grandiose narcissism and is comprised of admiration and rivalry dimensions. Admiration seeks greatness and popularity, while rivalry defends against conflict-related threats. Admiration and rivalry exhibit a moderate correlation, typically falling between 0.30 to 0.50 in observed correlations (*Back, 2018*). As for the relationship between rivalry and admiration, the relationship was positive but not strong. Thus, such findings support *Gauglitz et al. (2022)* conclusion: the correlation between the two dimensions of narcissistic admiration and narcissistic rivalry is moderate to strong. In other words, the two dimensions can exist together but are not always necessary.

The CFA fit indices showed an acceptable fit between the model and the collected data. Conducted CFAs have validated the two-dimensional structure, encompassing the admiration and rivalry factors. These dimensions each encompass affective-motivational, cognitive, and behavioral elements, as corroborated by studies conducted by *Back et al. (2013)*; *Rahmatpour, Peyrovi & Nia (2021)*; *Vecchione et al. (2018)*. The CFA results further supported the two-factor structure of the scale in the Algerian environment, demonstrating good accuracy and consistency. This sheds light on the fact that the scale is valid in the Arab environment. The Rasch measurement model results additionally demonstrated the likelihood of a person correctly responding to a specific item while considering the person's level of capability and the difficulty level of each item. Furthermore, the results of the Rasch model successfully arranged the scale items, resulting in a final set of 18 items after scaling. We also analyzed the category function and found that the six categories were monotonically increased with difficulties, indicating that the six categories are in order.

### Theoretical implications

The study validates the hypothetical model of NARQ, demonstrating its applicability in measuring narcissistic tendencies in the Algerian-Arab environment. It supports the two-dimensional structure of narcissism, specifically rivalry and admiration. The sub-scales

within each dimension exhibit strong correlations with their respective main dimensions, indicating robust relationships between the sub-dimensions and overall narcissism. Regarding the relationship between rivalry and admiration, the study found a positive but moderate association between the dimensions of rivalry and admiration of narcissism. This suggests that while these dimensions can coexist, they are not necessarily highly interrelated. This study thus contributes to measuring narcissism in the Arab-Algerian context by providing a valid model and scale within this cultural context.

## Practical implications

The present study aimed to examine the reliability and validity of the NARQ for assessing narcissism in the Algerian-Arab context. Utilizing this scale ensures more precise diagnosis and effective treatment planning in relation to narcissistic traits and behaviors. Furthermore, as part of this study, the NARQ was culturally adapted and translated to align with the Algerian-Arab environment. This adaptation provides practitioners with a measurement tool specifically tailored to the cultural context, enabling more accurate assessments and culturally appropriate interventions across Arab societies that share a common language. Finally, the psychometric investigation of the NARQ presents a dependable and valid instrument for assessing narcissism in the Algerian-Arab context. This carries significant practical implications, including precise assessment, culturally sensitive interventions, treatment monitoring, and further advancements in narcissism research.

## Limitations and recommendations for future research

Although the results obtained in this study are satisfactory, some limitations must be acknowledged regarding the NARQ's psychometric properties. First, the study sample consisted solely of university students, with the majority being females, which may limit the generalizability of the findings. It would have been more appropriate to include diverse participants, including patients, healthy individuals, and adults with a balanced gender ratio from various segments of society. Additionally, convenience sampling was employed, which is easy but may not be representative. Future studies should consider utilizing random sampling techniques to enhance the sample's representativeness. It is recommended that the scale be administered to different samples, especially patients, and that random sampling methods be employed in future research. Another limitation lies in our application of the scale to a single sample from Algeria. The scale might yield distinct results if a study is conducted across various Arab countries. Therefore, we recommend conducting cross-cultural studies in the future.

Additionally, the study was cross-sectional, and this is one of its significant limitations. Despite these limitations, the NARQ shows promise for future research. Its psychometric study in the Algerian-Arab environment contributes to the field of narcissistic research, and researchers and practitioners can explore additional properties of the scale, conduct further studies, and validate its applicability in diverse settings and populations. Continued research will advance our understanding of narcissism and its measurement. Finally, it is essential to address items with low loadings to improve the measure. For example,

the translation could be revised for poorly performing items, especially if these items are argued to be theoretically critical ("These items are essential for capturing the essence of grandiosity"). If these items are core to the construct, their low loadings suggest that the captured construct may not align with the intended one. Therefore, revisiting and refining these items is necessary to ensure the measure's validity and usefulness.

## CONCLUSION

The present study aimed to examine the psychometric properties of the NARQ following its translation into Arabic and using a sample of Algerian students. The results demonstrated satisfactory validity and reliability of the scale, as evidenced by CFA and Rasch analysis. By establishing the validity of the NARQ, this study has provided a comprehensive tool for measuring narcissism within the Arab-Algerian context.

### Funding
Ditmanson Medical Foundation Chia-Yi Christian Hospital supported the APC of the article. The funders had no role in study design, data collection and analysis, decision to publish, or preparation of the manuscript.

### Grant Disclosures
The following grant information was disclosed by the authors:
Ditmanson Medical Foundation Chia-Yi Christian Hospital.

### Competing Interests
The authors declare there are no competing interests.

### Author Contributions
- Aiche Sabah conceived and designed the experiments, performed the experiments, analyzed the data, prepared figures and/or tables, authored or reviewed drafts of the article, and approved the final draft.
- Musheer A. Aljaberi conceived and designed the experiments, performed the experiments, prepared figures and/or tables, authored or reviewed drafts of the article, and approved the final draft.
- Salima Hamouda performed the experiments, authored or reviewed drafts of the article, and approved the final draft.
- Djamila Benamour performed the experiments, authored or reviewed drafts of the article, and approved the final draft.
- Keltoum Gadja performed the experiments, authored or reviewed drafts of the article, and approved the final draft.
- Yu-Chen Lai performed the experiments, authored or reviewed drafts of the article, and approved the final draft.

- Chuan-Yin Fang performed the experiments, prepared figures and/or tables, authored or reviewed drafts of the article, and approved the final draft.
- Amira Mohammed Ali performed the experiments, authored or reviewed drafts of the article, and approved the final draft.
- Chung-Ying Lin analyzed the data, prepared figures and/or tables, authored or reviewed drafts of the article, and approved the final draft.

## Human Ethics

The following information was supplied relating to ethical approvals (i.e., approving body and any reference numbers):

The Department of Social Sciences, Faculty of Humanities and Social Sciences, Hassiba Ben Bouali University, Chlef, Algeria, granted Ethical approval to carry out the study within its facilities (Ethical Application Ref: 2023/gandm/19).

## Data Availability

The raw data is in the Supplemental Files.

## Supplemental Information

Supplemental information for this article can be found online at http://dx.doi.org/10.7717/peerj.17982#supplemental-information.

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
