# Peer review of "Psychometric characteristics of the Narcissistic Admiration and Rivalry Questionnaire (NARQ): Arabic version"

_PeerJ, doi:10.7717/peerj.17982_

## Round 0.1 · original submission · Major Revisions

Dear Dr. Aljaberi,

Thank you for submitting your manuscript to PeerJ. Your paper was evaluated by two individuals with considerable expertise in this area. Both reviewers are also an expert in quantitative psychology. Both have found your research question and study very interesting and have commented on a number of significant strengths. At the same time, they have identified several concerns. Based on my independent reading, I agree that your paper has potential for an impactful contribution, but some revisions are necessary. Thus, I am inviting you the option of revising and resubmitting it for consideration.

The reviewers have provided exceptionally thorough and constructive comments. I will not repeat their points but I urge you to carefully attend to their points, especially those related to the validity of the findings and your statistical analyses. You are welcome to submit a revision, if you think that you can address the concerns.

Thank you for giving us the opportunity to consider your work, and we wish you all the best with your article.

Yours sincerely,

Andree Hartanto
Academic Editor
PeerJ

Reviewer 1 ·

Basic reporting

Thank you for the chance to read this interesting psychometric paper. I have some major and minor comments which should be addressed. The overall reporting of the paper is generally clear and reasonably easy to follow. However, the argument made in the introduction (re: which model of narcissism is the most appropriate/useful one) is not strong enough. Please find more detailed comments below.

Introduction
1. From line 74, it is stated that narcissism is associated with "negative behavioral outcomes, including [...] decreased life satisfaction". Life satisfaction is not a behavioral outcome. Please either remove it or restructure this portion of the introduction.
2. From line 81, it is stated that narcissism is a "self-importance trait characterized by an exaggerated sense of worth and importance" which is quite repetitive/redundant. Please rephrase.
3. From line 89, the authors argue that narcissism "[shifts] between subclinical and clinical manifestations". While I don't necessarily disagree with this argument, it is not very well supported in the current introduction (e.g., there is no followup explanation, no lead up to this, and no citations provided). In addition, the authors are attempting to argue that this conclusion (re: subclinical and clinical shifts) follows from both the Krizan model and the Miller model. It is not clear why these conclusions have been drawn by the current authors from what has been explained about the two models.
4. In the "The Narcissistic Admiration and Rivalry Concept" section, yet another model is introduced. While there is an attempt to explain this model, it is not made clear why this model is appropriate/required/better as compared to the previous models which were already introduced. Please provide a clearer explanation and comparison.
5. The section detailing the validity of the NARQ in various populations is not well-structured. Please reorganise the section in a way that is easier for readers to follow, perhaps either by population or by what about the validity is exactly being examined.

Experimental design

Method
1. How did the authors ensure that the same participant did not complete the survey twice?
2. How was the sample size of 714 determined to be "sufficient" (line 185)?
3. In lines 212-213, please provide version numbers for all software used.
4. Please state if there was any missing data (e.g., participants skipped any questions) and if so, how it was handled.
5. Please additionally report TLI as part of the model fit indices.
6. The first paragraph of Data Analysis is very long; please consider breaking the paragraph for readability purposes.
7. In the Results section, in the Rasch Analysis portion, the authors draw conclusions based on distributions (first paragraph) and based on separation reliabilities/indices (second paragraph). Please provide some information about how such conclusions are drawn in the Data Analysis portion of the Method, for the benefit of readers who may not be familiar.

Validity of the findings

Results
1. In line 265, please report whether all factor loadings were statistically significant or not. Please state whether the factor loadings are standardised or unstandardised (they seem like they are standardised, but please be clear).
2. Please standardise the number of decimal places reported for skewness in table 2 and for difficulty and OUT.MNSQ in table 4. In addition, since skew and kurtosis are not bounded by [-1, 1], leading 0s should not be omitted.
3. Please report the four moments in the correct order (M, SD, skew, kurtosis) in table 2.
4. Factor loadings should be reported in table 2.

Additional comments

Please proofread the manuscript. There are minor typos such as "sqaure" instead of "square" (line 263). extra spaces (line 277), and inconsistent capitalisation (table 1).

Reviewer 2 ·

Basic reporting

The introduction does not adequately frame the research question within the current body of knowledge. There appears to be a disjointed narrative flow, and the relevance of cited studies is sometimes unclear which weakens the argument for the study's necessity.
o Several models of narcissism are discussed throughout the manuscript in different sections (line 81 discusses Krizan & Herlache (2018); line 85 discusses Miller et al. (2016); line 103 discusses Back et al. (2013)). It would be good if these could be collated together and evaluated to arrive at a conclusion on why the model by Beck et al. (2013) is taken as the focus in the current work, instead of adding them at different points of the manuscripts disjointedly.
o It’s unclear why the inclusion of the sentence in line 62-64 adds to your argument, and the sentence also seems out of place with the rest of the paragraph.
o Line 71-72 mentions “The factorial structure 72 of different measures of narcissism has revealed several aspects.” It would be good to elaborate further on these aspects.
o Line 128 states “This model has been subject to verification through seven studies”. Please cite the relevant studies here.

The language used in the manuscript lacks clarity in several areas, which could potentially confuse the reader. Specific phrases and sentences are convoluted, impacting the overall readability. I recommend a thorough revision for language and possibly enlisting a professional language editing service.
o The phrase ‘recent studies have explored narcissism” is repeated multiple times throughout the introduction (lines 48-49, line 71, line 79, line 101). It would be good to consolidate those mentions and create a better flow.
o “However” in line 67 is unnecessary.
o Line 94 states “narcissism's antagonistic and antagonistic aspects”. Is this a typo?

The manuscript would benefit significantly from the inclusion of the analysis scripts or syntax files used in SPSS, AMOS, and Winsteps. I recommend that the authors provide these files, clearly documenting each step of the data analysis process as outlined in the manuscript. This will allow other researchers to replicate the study's analysis more effectively and verify the findings independently.

Experimental design

The introduction currently emphasizes the increasing attention towards narcissism but lacks a clear justification for why the NARQ needs to be specifically validated in an Arabic-speaking Algerian sample. It would strengthen the manuscript to include a discussion on:
o The cultural specificity of narcissistic traits and how they might differ in Arabic cultures compared to those where NARQ was originally developed and validated.
o The potential clinical, social, and research implications of having a validated Arabic version of the NARQ, especially in terms of enhancing culturally appropriate assessments.
o Any existing gaps in the literature concerning the assessment of narcissism in Arabic-speaking populations and how this study aims to address these gaps.

Validity of the findings

The manuscript reports Cronbach's alpha values ranging from 0.59 to 0.84 for different subscales of the NARQ. While some of the values indicate acceptable to good reliability, the value of 0.59 is concerning as it falls below the commonly accepted threshold for acceptable reliability (α >= 0.70). This raises questions about the internal consistency of the subscale associated with this alpha value. It would be good to revaluate the items within this subscale to ensure that they are homogeneously contributing to the construct being measured, and consider revising or omitting items that might be lowering the subscale’s reliability. It would also be good to provide a discussion on the potential causes for this low alpha value and its implications for the scale's use in this cultural context.

The range of factor loadings from 0.48 to 0.84 raises some concern, particularly the items at the lower end of this spectrum which may not adequately represent the underlying construct. It would be beneficial for the authors to critically evaluate items with loadings below 0.70, as these may not contribute effectively to the construct they intend to measure. The authors can consider conducting item analysis to investigate whether these low-loading items are understood consistently by respondents or if they are ambiguously worded. It would also be good to provide a rationale for retaining any low-loading items, especially if they are crucial from a theoretical standpoint, and if not, consider revising or removing items with the lowest loadings to enhance the overall construct validity of the scale.

Additional comments

No Comments

---

## Round 0.2 · Minor Revisions

Dear Dr. Aljaberi,

Thank you for submitting your manuscript to PeerJ. We have just obtained the reviews from our experts. I also have read the manuscript myself independently before looking at the reviews. Overall, I am satisfied with the revision and agree that your paper has potential for an impactful contribution.

One of the reviewers also raised a number of relatively minor concerns that you should consider to address. Pending the revision, I am happy to conditionally accept your paper for publication in PeerJ

Thank you for giving us the opportunity to consider your work, and we wish you all the best with your article.

Yours sincerely,

Andree Hartanto
Academic Editor
PeerJ

Reviewer 1 ·

Basic reporting

The authors have attempted to address most of my and the other reviewer's comments, and have done so to a mostly satisfactory level in my opinion. However, I still have comments on the revised manuscript.

The manuscript reads much better now that it has been revised and proofread. However, it still needs to undergo a fresh round of proofreading due to the new additions to the text. For example, lines 324-327 "When all items are statistically significant, but some items have loadings less than 0.70, which is considered the threshold value for item loading onto factors, according to (Hair Jr et al., 2021). However, researchers have suggested a minimum loading value of 0.40 (Stevens, 2009)." are not grammatically correct at all. There are also extra spaces (e.g., "mean =0.00; SD =0.62" in line 352).

Experimental design

No comment

Validity of the findings

"skewness values ranged between -0.71 to 1.43" Please highlight the item(s) with skewness greater than 1 and plot their distribution. A skewness of >1 or <-1 implies non-normality, which will be problematic for SEM (incl. CFA).

"Cronbach's Alpha values for subscales showed acceptable values (0.59 to 0.84)". As the other reviewer has suggested previously, an alpha of .59 is not "acceptable". It is alright to obtain a value of .59 and then provide some argument for why the scale is still functional despite the low alpha value. But it is not accurate to interpret an alpha of .59 as "acceptable" by any conventional cutoffs. Please rephrase.

"The factor loading values ranged between 0.47 and 0.83, which are statistically significant at a significance level of 0.01." This sentence belongs under the CFA section. Factor loadings are derived from the CFA.

"Items with loadings below 0.70 were retained due to their theoretical importance in interpreting the latent variable." I don't disagree with this. However, more substantiation is required. Which items had loadings below .70, and how were they theoretically important to the latent variable? At least one example is necessary.

"Additionally, the minimum number of items in the latent variable should not be lower than three indicators or items, and deleting any item would result in the latent variable containing only two items, which disrupts its theoretical component and alters its property." This is not substantiated at all through further explanation nor any references.

"Moreover, in any translation or adaptation project of a scale to another context, lower reliability coefficients should be expected for subscales and composite measures of the instrument in the target language compared to the original language." I agree with this argument presented by the authors as an explanation for why alpha might be lower in the translated version. However, it is not sufficient to simply stop here. Why should we accept a poorer quality measure just because it is translated? Some attempt should be made to improve the measure, else it is not useful.

Additional comments

No comment

---

## Round 0.3 · Minor Revisions

Dear Dr. Aljaberi,

Thank you for submitting your manuscript to PeerJ. We appreciate your efforts in revising your work and addressing the previous comments. Upon further review, the reviewers have provided helpful suggestions for improvement, noting that the manuscript requires additional revisions to ensure clarity, accuracy, and overall quality. I agree that addressing those comments will significantly improve the quality of the manuscript.

For any comments that cannot be addressed without new data, consider toning down the argument and acknowledging the limitations in the discussion. Please don't worry; I would like to assure you that PeerJ evaluates articles based on scientific and methodological soundness, not on subjective determinations of 'impact,' 'novelty,' or 'interest.'

I believe that the current paper will be important and contribute to the field once the paper is published after addressing these comments. Thank you for giving us the opportunity to consider your work, and we wish you all the best with your article.

Yours sincerely,

Andree Hartanto
Academic Editor
PeerJ

Reviewer 1 ·

Basic reporting

Introduction
- There is a major issue in the literature review. The authors make the claim that "At the same time, experimental results have consistently shown that narcissism is associated with both positive attributes, such as charm and cheerfulness, as well as negative behavioral outcomes, including exploitative and manipulative behavior, arrogance, social rejection, conflicts, and decreased life satisfaction (Edelstein et al. 2012; Fehn & Schütz 2021; Jauk et al. 2021)." At least two of those three references are non-experimental in nature. The argument should be revised or the references should be changed.
- Because of the aforementioned, I would warrant that *all* references in the introduction be double-checked for accuracy.
- Please spell out 'narcissistic personality disorder' instead of using the NPD acronym.

Results
- The "the minimum number of items in the latent variable should not be lower than three indicators or items" claim is not supported by any references, and is objectively wrong. Two-item latent variables can be mathematically and empirically defined (e.g., see chapter 5 of Hoyle's Handbook of Structural Equation Modeling which states that latent variables should have at least two indicators, not at least three indicators). In fact, some quantitative/statistics researchers recommend the use of two-indicator latent variables (e.g., doi.org/10.1186/1471-2288-12-159).

General
- The paper requires proofreading (again). Examples (non-exhaustive):
-- There is an extra space before a comma in line 312.
-- "When all items are statistically significant, but some items have loadings less than 0.70, which is considered the threshold value for item loading onto factors, according to (Hair Jr et al. 2021)." is not grammatical.
-- "Items with loadings below 0.70 were retained due to their theoretical importance in interpreting the latent variable. Despite their loadings being below 0.70, certain items are theoretically crucial and cannot be removed." is repetitive.

Experimental design

See next section

Validity of the findings

In the introduction, it is explicitly stated that "This study aims to provide a valid and reliable tool for measuring narcissism in the Algerian environment using the NARQ". However, this aim was not fully met, since the current measure obtained reliability coefficients etc of which some were subpar.

As stated in the previous round of comments: Why should we accept a poorer quality measure just because it is translated? Some attempt should be made to improve the measure, else it is not useful. For example, perhaps the translation could be altered for the items that are not performing well. This is especially so if the items with poor performance are those that the authors argue are theoretically critical ("These items are essential for capturing the essence of grandiosity"). If the items are so core to the construct that they cannot be dropped, why are the loadings so low? This implies that the construct being captured may NOT be what is actually intended to be captured. Factor loadings do directly influence what construct is being captured (e.g., see doi.org/10.1186/1471-2288-12-159).

Simply stating that "the NARQ translated into the Arabic for the present study should be refined in future studies. In other words, the present study provides a beginning of the Arabic version of NARQ for future studies to further investigate" is not a very strong argument. It seems more to me like the attempt to develop a "valid and reliable [translated] tool" was not successful, and some attempt should be made to move closer to a successful outcome (e.g., trying a different translation of the problematic items). I do not find it very meaningful to know that an attempt was made to translate it, and the measure did not perform up to standard. This paper showed much promise, if only it were completed and produced a high quality measure that others could use.

Additional comments

No comment

Reviewer 2 ·

Basic reporting

This is a work I have reviewed previously, aiming to validate the Arabic translation of the NARQ among Algerian university students. I thank the authors for their responsiveness to the comments. While the manuscript is much improved, there are still some concerns.

Please proofread the paper again to check for grammar, general clarity and flow.
- Line 310 – 313 “When all items are statistically significant, but some items have loadings less than 0.70, which is considered the threshold value for item loading onto factors, according to (Hair Jr et al. 2021). However, researchers have suggested a minimum loading value of 0.40 (Stevens 2009).” This section is not grammatical at all, and needs to be rephrased.
- Line 81 – 83 “Despite their diversity, there has 82 been a recent surge in narcissism studies, leading to a better understanding of the concept (Sivanathan et al. 2021a).”, the sentence seems out of place.
- Line 91 – 92 “Despite variations, they offered similar definitions, suggesting a spectrum from subclinical to clinical manifestations.”, offered should be changed to offer.

I suggest the authors provide a more detailed explanation of why the Arabic version of the NARQ is necessary. For example, discuss the cultural shifts in Arabic societies from collectivism to individualism and how these changes might impact the expression of narcissistic traits. Currently, the explanation provided (line 162 – 164) states that "Prior evidence shows that collectivist individuals, compared to individualist individuals, score higher narcissistic scores (Foster et al. 2003)." This seems contradictory, as the previous sentence mentions that Arabic cultures have shifted to individualistic. The authors need to clarify this point and explain the relevance of an Arabic version of the NARQ in light of these cultural shifts. How does this shift impact the necessity and applicability of the NARQ in Arabic-speaking populations?

Experimental design

In lines 221-223, the manuscript states, "Each strategy consists of three subscales (three items each) designed to measure narcissism's behavioral, emotional, motivational, and cognitive dynamics." However, the sentence mentions four dynamics (behavioral, emotional, motivational, and cognitive) but only three subscales. This discrepancy needs to be addressed.

Lines 307-308 state "All other fit indices support a good fit except for the TLI, which is close to acceptable." It would be more informative to provide the exact TLI value. This transparency allows readers to better assess the model fit. Please include the specific TLI value to clarify how close it is to the acceptable threshold.

Validity of the findings

In lines 316-317, the manuscript states, "Items with loadings below 0.70 were retained due to their theoretical importance in interpreting the latent variable." However, the factor loadings for these items range from 0.40 to 0.70, which is relatively low. This raises a concern: if these items are indeed theoretically important, why do they exhibit such low factor loadings? For instance, the low factor loadings might indicate issues with the translation or cultural adaptation of the items.

Lines 343-348 state, "Moreover, in any translation or adaptation project of a scale to another context, lower reliability coefficients should be expected for subscales and composite measures of the instrument in the target language compared to the original language. In this regard, the NARQ translated into Arabic for the present study should be refined in future studies. In other words, the present study provides a beginning of the Arabic version of NARQ for future studies to further investigate." While it is true that translations can result in lower reliability coefficients, this explanation is not sufficient to justify the acceptance of the Arabic version with lower reliability. Simply stating that future studies should refine the scale does not adequately address the current study's effort to create a usable scale. The authors need to provide a stronger justification for why the current version should be accepted despite its lower reliability. This could include discussing specific steps taken to ensure the quality of the translation, the potential impact of the lower reliability on the scale's validity, and how the current version can still be useful in its present form.

In lines 418-419, the manuscript states, "The scale demonstrated high levels of internal consistency." This assertion appears to be inconsistent with earlier sections of the manuscript that discuss lower reliability coefficients for the translated version. The authors should clarify this point. If the internal consistency is not as high as claimed, it would be more accurate to discuss the actual reliability coefficients obtained and provide a nuanced interpretation of what these findings imply for the use of the scale in its current form. This could include acknowledging any limitations and explaining how the scale might still be useful, while emphasizing the need for further refinement and validation.

---

## Round 0.4 · accepted · Accept

Dear Dr. Aljaberi,

Thank you for your rigorous revision addressing the comments from both reviewers. Although I attempted to send the revised manuscript to the reviewers for a final check, it appears that both are currently unavailable.

Nevertheless, I have independently reviewed the revised manuscript and your responses to the reviewers' comments carefully. I appreciate the extensive efforts that you and your team have made to improve the quality of the manuscript. The manuscript is now stronger, with solid methodological justification, improved writing, good implications, and a careful discussion of the study's findings.

I am pleased to inform you that the above paper has been accepted for publication in PeerJ. Thank you for giving the Journal the opportunity to publish your work. I believe it will make a significant contribution to the literature. Well done!

Best regards,

Andree